# Cortical Auditory Evoked Potentials in Children with Prenatal Exposure to Zika Virus

**DOI:** 10.3390/v14091923

**Published:** 2022-08-30

**Authors:** Laís Cristine Delgado da Hora, Lilian Ferreira Muniz, Leonardo Gleygson Angelo Venâncio, Karina Paes Advíncula, Jéssica Dayane da Silva, Diana Babini Lapa de Albuquerque Britto, Demócrito de Barros Miranda Filho, Elizabeth B. Brickley, Ricardo Arraes de Alencar Ximenes, Silvio da Silva Caldas Neto, Mariana de Carvalho Leal

**Affiliations:** 1Department of Speech, Language and Audiology, Universidade Federal de Pernambuco, Recife 50740-520, Brazil; 2Graduate, Program in Human Communication Health, Universidade Federal de Pernambuco, Recife 50740-520, Brazil; 3Pós-Graduação em Ciências da Saúde, Universidade de Pernambuco, Recife 50100-010, Brazil; 4Department of Infectious Disease Epidemiology, London School of Hygiene & Tropical Medicine, London WC1E 7HT, UK; 5Pós-Graduação em Medicina Tropical, Universidade Federal de Pernambuco, Recife 50670-901, Brazil; 6Department of Surgery, Universidade Federal de Pernambuco, Recife 50740-520, Brazil

**Keywords:** Zika virus, congenital infection, Congenital Zika Syndrome, microcephaly, Evoked Potentials, auditory, electrophysiology, auditory cortex, hearing, child development

## Abstract

Prenatal exposure to ZIKV can cause neurologic and auditory damage. The electrophysiological responses obtained by Cortical Auditory Evoked Potentials (CAEP) may provide an objective method to investigate the function of cortical auditory pathways in children exposed to ZIKV. This case series analyzed the findings of CAEP in prenatal-period ZIKV-exposed children with and without microcephaly. The CAEP was performed in a total of 24 children. Five magnetic resonance imaging (MRI) images of the inner ear and brain of microcephalic children were analyzed and compared with CAEP measurements. Ventriculomegaly (80%), cortical/subcortical calcification (80%), and brain reduction (60%) were the most common alterations in the MRI. The P1-N1-P2 complex of the CAEP was observed in all children evaluated. The peak N2 was absent in two children. In the comparison of the CAEP measurements between the groups, children with microcephaly presented a higher amplitude of P2 (*p* = 0.017), which may reflect immaturity of the auditory pathways. Microcephalic and normocephalic children with prenatal exposure to ZIKV presented with the mandatory components of the CAEPs, regardless of changes in the CNS, suggesting that this population has, to some extent, the cortical ability to process sound stimuli preserved.

## 1. Introduction

Zika virus (ZIKV) is an arthropod-borne virus (arbovirus) of the *Flaviviridae* family, which was first isolated from humans in 1952. Between 2015 and 2017, a ZIKV epidemic occurred in Brazil, during which cases of congenital abnormalities were observed among neonates whose mothers were infected with the virus during pregnancy [1]. Since then, published research has cast light on the potentially neurotropic profile of ZIKV, whose targets include neural progenitor cells and neuronal cells that can be affected across all stages of the central nervous system (CNS) [2,3].

Today, Congenital Zika Syndrome (CZS) is recognized as a spectrum of clinical manifestations including both structural anomalies in different areas of the CNS, with microcephaly being one of the most prominent [2,4,5,6,7], as well as functional alterations affecting pediatric neurodevelopment and behavior [3,8,9,10].

Notably, there is some evidence to suggest that prenatally ZIKV-exposed normocephalic children, without evident changes at birth, can also present with long-term neurodevelopmental sequels, including delays in cognitive and language development [8,11,12]. However, some recent studies have observed similar rates of neurodevelopment delays in ZIKV-exposed normocephalic children when compared to a parallel group of socio-demographically matched unexposed controls [13,14]

Since 2019, the Joint Committee on Infant Hearing has added ZIKV to the list of prenatal infections that may increase risks of hearing loss and provided guidance recommending neonatal screening, preferably through brainstem auditory evoked potential (AEP), as well as the longer-term auditory follow-up of exposed children [15]. Nevertheless, knowledge about the long-term impacts of prenatal exposure to ZIKV on the auditory system still has gaps, especially related to the development of central auditory pathways and the skills involved in sound processing [16]. 

It is well established that the integrity of the auditory system is fundamental for the development of auditory abilities that occur in the first years of life. Cortical malformations and neurodevelopmental disorders can influence the maturation of the auditory pathways, impacting the processing of auditory information, language, and communication [17,18,19]. Cortical Auditory Evoked Potentials (CAEP) are long latency potentials that help in the diagnosis and monitoring of auditory development in the infant population [20]. They reflect the activity of the auditory cortex, providing information about the biological processes involved in the processing of sound information. In addition, they are used to evaluate auditory maturation, hearing capacity, and speech hearing in children with and without hearing loss [21].

In this context, the aim of this study was to analyze the findings of CAEP in prenatally ZIKV-exposed children with and without microcephaly. Studying electrophysiological responses obtained through CAEP can provide an objective method for investigating the function of cortical auditory pathways in children exposed to ZIKV.

## 2. Materials and Methods

This case series study with comparisons between children with and without microcephaly is linked to two major research projects entitled “Audiological Evaluation of Children with Suspected Congenital Zika Virus Infection” and the “Microcephaly Epidemic Research Group Pediatric Cohort (MERG–PC)” [22]. The research was carried out in an outpatient public health unit in the metropolitan region of Recife, Pernambuco, Brazil. The sample consisted of children of both sexes with confirmed and probable prenatal ZIKV exposure who were born between 2015 and 2016 (i.e., the period including and immediately following the peak of the ZIKV epidemic in the state of Pernambuco) and regularly followed up over a period of three years, with the aim of monitoring hearing. 

Ethical approval for the research project was provided by the Research Ethics Committees of the Hospital Agamenon Magalhães (CAAE 3.072.730) and the Faculdade de Ciências Médicas da Universidade de Pernambuco (CAAE 52803316.8.0000.5192). The parents and guardians responsible for the participants were informed about the study objectives and procedures and signed free and informed consent forms prior to participation. 

In this study, we defined confirmed cases as children and mothers who tested positive for ZIKV. The samples were tested for Zika virus-specific IgM antibodies using a capture ELISA based on the US Centers for Disease Control and Prevention (CDC) Emergency Use Authorization protocol, with reagents from the CDC (Fort Collins, CO, USA) [23]. The presence of Zika virus specific neutralizing antibodies was assessed in the serum samples of mothers and neonates by the plaque reduction neutralization test (PRNT50), with a 50% cut-off value for positivity and immunoglobulin. 

We defined probable cases as children with no laboratory evidence ZIKV testing results based on neonatal samples whose mothers presented rash and/or fever during pregnancy (i.e., with no definitive alternative cause) but with laboratory evidence (i.e., by a combination of qRT-PCR, IgM, IgG3, and/or plaque reduction neutralization test (PRNT50)) of ZIKV infection during pregnancy. 

This case series included children who presented with no evidence of peripheral hearing alterations, as indicated by the presence of wave V in the newborn hearing screening test and during auditory monitoring in the prior year using BAEP with click stimulus at the intensity of 35 dB nHL. This study excluded children with (i) congenital infections with other viruses, (ii) other congenital syndromes, (iii) known exposure to risk factors for hearing alterations according to the 2019 Joint Committee on Infant Hearing, and/or (iv) signs of alteration in peripheral auditory pathways [15].

The participants were categorized in two groups based on head circumference at the time of birth. The microcephalic group included children with a head circumference Z-score of >2 standard deviations below the mean for age and sex, according to InterGrowth-21st reference curves [24]. The normocephalic group included children with a head circumference Z-score within the normality range (i.e., ≤2 standard deviations below to ≤2 standard deviations above the mean for age and sex) [24].

Among the children with microcephaly, five underwent magnetic resonance imaging (MRI) of the inner ear and brain based on clinical indication. The MRI analysis was performed independently by two examiners, an experienced neuroradiologist and an experienced otorhinolaryngologist, and blinded in relation to the clinical presentation of the participants. Cases of disagreement were discussed together and decided by consensus. Imaging results were evaluated for abnormalities in the inferior and temporal frontal cerebral parenchyma, ventricles, brainstem, thalamus, callous body, cerebellum, cochlea, labyrinth, and cochlear and vestibular nerves (Figure 1). Clinical outcomes of interest included the presence of hypoplasias, hydrocephalus, calcifications, and ventriculomegaly (Table 1).

On the day of the auditory evaluation, an audiological screening was performed with otoscopy and imitanciometry (Interacoustics equipment, model AT-235), to exclude external and middle ear alteration. The normality pattern adopted was type A curve (admittance peak 0.3 to 1.6 mL and peak pressure −100 to +100 daPa) and presence of ipsi and contralateral stapedial reflexes. Participants were then evaluated for the CAEP examination, which was performed using Intelligent Hearing Systems (IHS) equipment. Four copper electrodes were used. The ground electrode was positioned in Fpz, the non-inverting electrode in Fz, and two inverting electrodes in the right (A1) and left (A2) lobes.

During the CAEP examination, the children were positioned in a reclining armchair and remained awake in a state of stillness watching a video on an electronic tablet device without sound. Children lacking postural stability were positioned in the lap of their responsible parent or guardian.

Pure tones of 1000 Hz and 4000 Hz to 80 dB HL via insertion phones (ER-3) were presented in a binaural and random way, with the proportions including 20% rare stimuli (4000 Hz) and 80% frequent stimuli (1000 Hz). The acquisition parameters were: impedance maintained at 1 kOhms, recording window of 510, filtering passes band of 1–30 Hz, alternating polarity, stimulation rate of 1.1 stimulus per second, and total of 500 sweeps.

The CAEP were recorded and analyzed by two trained evaluators with experience in electrophysiology. The identification of the waves of the examination was performed in the tracing formed by the rare stimulus. For analysis, the presence of components P1, N1, P2, and N2, as well as their latencies and amplitudes, were identified. Any disagreements were resolved by consensus.

All statistical analyses were performed using *Statistical Package for the Social Sciences* (SPSS), version 21.1, software (New York, NY, USA), and the significance level adopted was 5%. Data normality was verified with the Shapiro–Wilk test. Depending on the distribution of the data, the CAEP latency and amplitude measurements were compared between groups using independent t-tests and Mann–Whitney U tests. The point-biserial correlation coefficient was applied to verify the existence of correlation between the measurements of the CAEP and the findings of the MRI results in the children with microcephaly.

## 3. Results

In total, this study evaluated 24 children and analyzed the data from 21 children, of whom 11 were microcephalic and 10 were normocephalic. Three children were excluded from the final sample because their tests were considered to contain artefacts, and it was not possible to interpret their results as reliable electrophysiological responses.

Among the children with microcephaly included in the study, six were female and five were male, with a mean age of 45.6 months (standard deviation of 2.2 months). Four of the participants had severe microcephaly, with a head circumference of >3 standard deviations below the mean for age and sex, according to the InterGrowth-21st reference curves [24]. In the group of normocephalic children, four were female and six were male, with a mean age of 39.1 months (standard deviation of 4.0 months).

Table 1 shows the frequency of alterations of the five imaging exams of children with microcephaly. The following alterations were found when the brain MRI was evaluated: ventriculomegaly (80%), cortical/subcortical calcification (80%), and brain reduction (60%).

In the CAEP measurements, a single outlier finding was observed in one of the children with microcephaly (Table 1, S04, P2 = 14.86) that did not follow the common distribution of amplitude measurements. Once there was control over the measurement and data entry, it was decided to keep this finding.

The frequencies, means, standard deviation, minimum and maximum interval, and significance value of the components of the CAEP in children exposed to ZIKV with and without microcephaly are described in Table 2. There was no significant difference in the comparison between the mean latencies of the CAEP in all components and the amplitude means of the P1, N1, and N2 components of the groups with and without microcephaly. However, children with microcephaly presented with greater P2 wave amplitudes than the group of children without microcephaly (*p* = 0.017).

In comparing the amplitude findings between the groups, a statistical analysis was performed without the previously described outlier finding. It was observed that the means of the groups are different and, on average, children with microcephaly had a higher mean amplitude of P2 than children without microcephaly (t(18) = 2.34; *p* = 0.31). We performed the Mann–Whitney test to confirm our hypothesis and observed again that the presence of microcephaly had an effect on the means of the groups, so that the groups were statistically different (U = 21.0; *p* = 0.028). Therefore, when comparing the results with and without the presence of the outlier by means of statistical tests appropriate to the variance of the data, the statistical difference between the groups was maintained.

Table 3 shows the correlation coefficient between imaging findings, latency, and amplitude of the CAEP in the group of children with microcephaly. Within this small sample, there appeared to be a negative association between the occurrence of changes in the brainstem and thalamus and the amplitude of N2. 

## 4. Discussion

In this study, the P1-N1-P2 complex of the CAEP was observed in all children evaluated, regardless of the presence of microcephaly; however, peak N2 was absent in two children (one with microcephaly and the other normocephalic). In the comparison of the CAEP measurements between the groups, children with microcephaly presented a higher amplitude of P2. There was also some evidence to suggest an association between the measures of the CAEP and the MRI findings. 

The evaluation of peripheral auditory pathways in children diagnosed with confirmed or probable congenital ZIKV infection, as well as the language and auditory behavior reported by parents, have been described in the literature [25,26,27,28]. However, up to the time of this article, no studies have been found that have used CAEP to evaluate the central auditory pathway in children with prenatal exposure to ZIKV. 

The current evidence base in the published literature remains insufficient to fully understand the impacts caused by ZIKV on the central auditory system [29]. The largest series of cases published on hearing assessment in children with CZS reported that there was prevalence of 9.3% of peripheral auditory alterations; however, no alterations with late onset or progressive profiles was found. This finding, however, does not discount the fact that the central auditory system is less susceptible to alterations, because cortical responses depend on the integrity of the bottom-up and top-down pathways, in addition to factors such as age and auditory experiences [27].

The P1-N1-P2 complex of the CAEP was observed in all children evaluated. This complex has been present since childhood and indicates neural response in the auditory cortex in the face of auditory stimulation [30,31]. From a neuroanatomical point of view, such findings are favorable, because these exogenous components involve the participation of important structures of cortical auditory processing such as the Heschl gyrus, primary temporal cortex and lemniscus, reticular formation, and thalamic and brainstem regions [32,33,34]. It is noteworthy that the presence of this complex requires less cognitive complexity, because they are mandatory auditory evoked responses related to stimulus characteristics. They are recorded invariably without taking into account the attention to stimuli, that is, they do not depend on the conscious attention of the individual for the conduction of information along the areas of association and auditory cortex [35].

In this sense, the neurological damage associated with exposure to ZIKV during fetal development did not appear to interfere with the occurrence of CAEP waves. The main explanation stems from the fact that the P1-N1-P2 complex primarily reflects the characteristics of the acoustic stimulus, although it depends on the integrity of the afferent cortical auditory pathways. Therefore, it cannot be ruled out that children exposed to ZIKV perceive the spectral and time characteristics, and also passively discriminate the auditory stimulus.

Peak N2 was absent in only two children, one with microcephaly and the other normocephalic. It is a mixed component that originates in multiple s-generated responses along the brain stem, cortical, and thalamic auditory regions [32,33], which may undergo alterations due to intrinsic factors such as attention and sleep [36]. Maturational aspects, including the individual variation in development intrinsic to each subject, and exposure to sounds in the first years of life, may also explain the absence of this peak [37]. Thus, it seems unlikely that this finding is directly associated with the occurrence of congenital ZIKV infection.

In the comparison of the measures of the CAEP between the groups, children with microcephaly presented with greater amplitude findings for P2 than their normocephalic peers. Amplitude measurements are directly related to the magnitude of neural response and the amount of nerve fibers recruited during the passage of the stimulus in the auditory [35]. The P2 component originates in several regions of the primary, secondary, and reticular auditory cortex, associated with the attention that the individual gives to the sound stimulus and with the inhibition of the processing of competitive stimuli. As the auditory pathways mature, the subjects need less auditory attention to discriminate the stimuli, adapt to them, or ignore them, which generates a wave of smaller amplitude [20]. Thus, it is possible that the greater amplitude of P2 in children with microcephaly is associated with immaturity of the auditory pathways in view of the impacts generated by the multiple structural alterations of the CNS and in the development of supra-auditory domains, demanding a greater effort and greater propagation of neural activation to discriminate the stimulus, increasing the amplitude of the wave.

The measurements of latency and amplitude of P1-N1-P2-N2 found in the children of both groups of this study are similar to those reported in another study that evaluated children from a similar age group who were born at term and presented with no auditory complaints or developmental alterations [20]. As previously mentioned, amplitude measurements are related to the magnitude and synchrony of the neural responses involved in acoustic stimulus processing. On the other hand, latency measurements are also related to the time of neural conduction of this stimulus in the auditory pathways [35]. As P1 can be considered a biomarker for investigating the development of cortical auditory pathways due to well-established latency and amplitude measures [21], such findings suggest that children prenatally exposed to ZIKV, even with microcephaly, have a pattern of maturation of cortical auditory pathways that are to some extent compatible with that of hearing children. Thus, it is likely that the beginning of cortical processing will occur, but it does not mean that it occurs with the same efficiency as the pattern of decoding for children without peripheral auditory alterations and developmental alterations. 

It is noteworthy that, while crude latency and amplitude values are an indication for the evaluation of central auditory pathways, they are not able to demonstrate the morphological differences in the electrical activity pattern of the children evaluated.

Another factor that draws attention refers to the association between the measurements of the CAEP with the findings of imaging tests in which it was observed that the presence of trunk and thalamus hypoplasia was associated with a greater amplitude of the N2 wave compared to the group without hypoplasia. It is known that this component is related to the ability to identify and care for the rare stimulus, being correlated to the difficulty level of the task [38]. Thus, the decrease in N2 amplitude, even in the absence of changes in the thalamus and brainstem, can be explained by a possible maturational slowness that would produce difficulty in passive and automatic sensory processing. Although this correlation has been observed from a statistical point of view, these findings should be interpreted with caution due to the restricted sample size. Further studies with larger samples are necessary to confirm the results.

In addition, the influence of individual variations intrinsic to the development of each individual cannot be ruled out, as well as exposure to different acoustic stimuli in the first moments of life. For example, a significant reduction in N2 amplitude and latency is reported for individuals with mild to moderately severe sensorineural hearing loss, a fact associated with sensory input problem [39,40] deficits in cortical synchronization [31]. 

Another important factor is that the structural alterations of the CNS present in children with microcephaly in this study may be associated with alterations in multiple cognitive domains, such as memory, attention, orientation, perception, language, executive functions, or information processing [25,41], which may contribute to the appearance of N2 amplitudes with lower values.

The presence of the mandatory components of the CAEP in this study suggests no change in the functioning of the central auditory pathway to the region of the generating sites of the N2 component. Due to the sample size and methodological clippings used, it is suggested that research with different designs can be carried out to better elucidate the efficiency of cortical decoding of sound stimuli in children prenatally exposed to ZIKV.

## 5. Conclusions

In this case series, we observed that microcephalic and normocephalic children with prenatal exposure to ZIKV presented with the mandatory components (P1-N1-P2-N2) of the CAEPs, regardless of changes in the CNS, suggesting that this population has, to some extent, the cortical ability to process sound stimuli preserved. However, the presence of microcephaly seems to interfere with the amplitude of peak P2, a component closely related to attention capacity and auditory discrimination. The presence of trunk and thalamus hypoplasia was associated with a greater amplitude of the N2 wave compared to the group without hypoplasia.

Future studies with larger samples, longitudinal follow-up, and use of verbal stimuli in the CAEP will provide more accurate information regarding the development and performance of the central auditory system for speech sounds at different times in development.

## Figures and Tables

**Figure 1 viruses-14-01923-f001:**
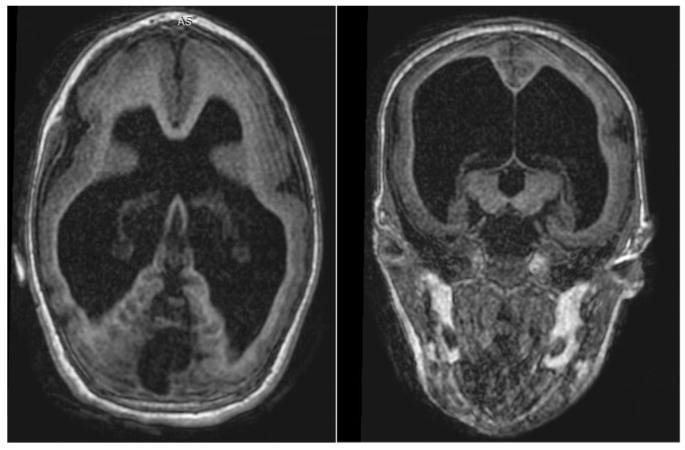
Axial and coronal sections of the magnetic resonance imaging (MRI) of one microcephalic child from this study.

**Table 1 viruses-14-01923-t001:** Description of age and severity of microcephaly as a measurement and frequency of changes in imaging of five children with microcephaly (*n* = 5).

	**S01**	**S02**	**S03**	**S04**	**S05**
Severe Microcephaly *			x		x
Age (months)	36	46	50	47	47
**Cortical Auditory Evoked Potentials measures**					
Latency (milliseconds)					
P1	119	78	111	48	90
N1	164	138	158	104	116
P2	205	195	206	292	161
N2	279	265	248	315	226
Amplitude (microvolt)					
P1	3.87	3.31	3.84	3.62	3.47
N1	2.72	3.06	3.72	7.95	2.07
P2	2.68	5.37	2.61	14.86	5.02
N2	1.92	14.17	1.11	2.82	8.26
**Magnetic Resonance Imaging findings**					
Hypoplasia in the Brainstem		x			x
Hypoplasia in the Thalamus		x			x
Cerebellar Hypoplasia	x				x
Hydrocephalus				x	x
Frontal Hypoplasia	x	x			
Temporal Hypoplasia	x	x			
Cortical/Subcortical Calcification					x
Cortical Malformation	x	x			
Ventriculomegaly	x	x	x	x	
Brain Reduction			x	x	x
Hypoplasia in Corpus Callosum			x	x	
Normal Internal Ear	x	x	x	x	x

* Severe microcephaly: head circumference Z-score of >3 standard deviations below the mean for age and sex, according to InterGrowth-21st reference curves [24].

**Table 2 viruses-14-01923-t002:** Results of comparing the measures of latency and amplitude of the CAEP in children prenatally exposed to ZIKV with microcephaly (*n* = 11) and without microcephaly (*n* = 10).

	Groups	%	Mean	Standard Deviation	Confidence Interval-Upper	Confidence Interval-Lower	*p*-Value
P1 Latency (ms)	Microcephalic	100%	101.5	31.1	48.0	153.0	0.428 ^1^
	Normocephalic	100%	91.9	22.1	51.0	122.0	
N1 Latency (ms)	Microcephalic	100%	157.7	46.2	96.0	246.0	0.776 ^1^
	Normocephalic	100%	152.5	35.4	85.0	194.0	
P2 latency (ms)	Microcephalic	100%	231.5	49.8	161.0	340.0	0.934 ^1^
	Normocephalic	100%	233.5	56.5	132.0	308.0	
N2 latency (ms)	Microcephalic	91%	278.0	38.9	224.0	330.0	0.421 ^1^
	Normocephalic	90%	294.3	47.2	236.0	365.0	
P1 Amplitude (μV)	Microcephalic	100%	6.56	3.11	2.1	10.1	0.157 ^1^
	Normocephalic	100%	4.33	2.28	1.1	7.4	
N1 Amplitude (μV)	Microcephalic	100%	6.78	4.71	1.9	14.9	0.205 ^2^
	Normocephalic	100%	3.89	1.74	0.5	7.3	
P2 Amplitude (μV)	Microcephalic	100%	6.15	4.06	1.1	14.2	0.017 ^2,^*
	Normocephalic	100%	2.50	2.04	0.1	6.9	
N2 Amplitude (μV)	Microcephalic	91%	4.85	3.47	1.0	12.4	0.286 ^1^
	Normocephalic	90%	3.36	2.21	0.4	7.4	

^1^ = Independent T test; ^2^ = Test Mann–Whitney; * = *p* ≤ 0.05; ms = milliseconds; μV = microvolts; % = peak presence frequency.

**Table 3 viruses-14-01923-t003:** Results of the point-biserial correlation coefficient between measurements of the CAEP and the findings of the MRI results in the children with microcephaly.

	Latency (ms)				Amplitude (µV)			
	P1	N1	P2	N2	P1	N1	P2	N2
Hypoplasia in the Brainstem	−0.14	0.36	0.36	0.12	0.10	−0.21	−0.78	−0.97 **
Hypoplasia in the Thalamus	−0.14	0.36	0.36	0.12	0.10	−0.21	−0.78	−0.97 **
Cerebellar Hypoplasia	−0.25	−0.35	0.63	0.48	0.43	0.60	−0.70	−0.33
Hydrocephalus	0.80	0.81	0.02	0.21	0.31	−0.39	−0.36	−0.30
Frontal Hypoplasia	−0.45	−0.66	−0.23	−0.56	−0.66	0.02	−0.20	−0.16
Temporal Hypoplasia	−0.45	−0.66	−0.23	−0.56	−0.66	0.02	−0.20	−0.16
Calcification cortical/subcortical	0.03	0.41	0.75	0.71	0.73	0.22	−0.78	−0.69
Cortical Malformation	−0.45	−0.66	−0.23	−0.56	−0.66	0.02	−0.20	−0.16
Ventriculomegaly	−0.03	−0.41	−0.75	−0.71	−0.73	−0.22	0.78	0.69
Brain Reduction	0.45	0.66	0.23	0.56	0.66	−0.02	0.20	0.16
Hypoplasia in Corpus Callosum	0.42	0.32	−0.38	−0.01	0.06	−0.20	0.84	0.73

** *p* < 0.001.

## Data Availability

Data cannot be shared publicly because public availability would compromise patient privacy. De-identified data can be made available upon reasonable request from qualified investigators by contacting the Laboratório de Audiologia do Departamento de Fonoaudiologia da Universidade Federal da Pernambuco (UFPE) at depot.fono@ufpe.br.

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
