# Peer review of "Cortical Auditory Evoked Potentials in Children with Prenatal Exposure to Zika Virus"

_viruses, 2022, doi:10.3390/v14091923_

Round 1

Reviewer 1 Report

This is an interesting and well-written study that presents cortical auditory evoked potential findings from a rare sample of children with laboratory evidence of prenatal Zika virus exposure. I have only a few concerns about how the data is presented and whether findings are sufficiently supported by the data.

In the introduction, there is no mention of studies that have emerged in the past few users showing similar rates of neurodevelopment delays in ZIKV-exposed normocephalic children when compared to a parallel group of socio-demographically matched unexposed controls (PMID: 31878830 and PMID: 34479857). These studies are especially important to mention given that the predominant message from the current study appears to be that ZIKV-exposed children show broadly normal mandatory components of the CAEPs.

In the methods section, the sentence explaining criteria for probable cases is difficult to understand.There may be a key word missing. Should it read “no laboratory evidence…”??? 

In the results section, my main concern is that a single outlier finding in one microcephalic child (Table 1, S04, P2=14.86) may be driving the P2 amplitude finding. It is a small sample and a single outlier can have a more significant impact. Does the finding remain when S04 is removed? Could an outlier like this be due to artifact? 

In the discussion section, the presentation of a relationship between lower P2 amplitude and maturation of the auditory pathways is interesting. If the P2 finding is not being driven by a single outlier, then this interpretation (that higher P2 amplitude may reflect maturational delay) should be briefly mentioned in the abstract. It makes the finding more meaningful to those who do not do CAEP studies.

In the discussion, there is mention of a normative P1-Ni-P2-N2 study [18]. What are the base rates of abnormally elevated P2 amplitude in normal children? How is this typically interpreted and handled when it occurs in normal children? Is it excluded as an outlier? A similar approach should be applied in the current clinical sample. Or if not (because each child is valuable and worth including), then the results with and without the outlier included should be presented so the results can be compared to prior normative studies.

Discussion of the findings from the CAEP and MRI correlations should be couched in more caveats and limitations given the small sample and possible influence of outliers.

Minor revisions

-10th line of abstract “should be “children with microcephaly”

-3rd paragraph of discussion, second sentence is difficult to understand and may need to be re-written for clarity. Are peripheral auditory alterations present in children with CZS or not?

Author Response

Dear Reviewer,

We appreciate your comments and suggestions. The questions received were answered in the annex.

Kind regards,

Lais Delgado

Reviewer 2 Report

Zika virus is a mosquito-borne flavivirus. Unlike other flaviviruses, Zika virus causes congenital neurological abnormality in neonates when mothers experienced Zika virus infection during pregnancy. Authors analyzed CAEP to understand effects of prenatal Zika virus exposure on the development of cortical auditory functions, which could contribute to understand pathogenicity of Zika virus. Several points were raised.

1.     Materials and Methods: Regarding to ELISA, did authors used Kit? or in house ELISA? Please clarify and provide the info.

2.     Materials and Methods: Please describe cell line used for neutralization test.

3.     Table 1: How did authors define “severe microcephaly” or not?

4.     Table 1: The values for P1 seem to be quite high (387, 331, 384, 362, 347). Are these correct?

5.     Table 2: What does “% column” show?

6.     Table 2: Why did authors utilize different statistical analysis only for N1 and P2? Are these based on their variance?

7.     Table 2: Is it possible to provide the data obtained from healthy control unexposed with Zika virus?

Author Response

(The authors gave the same response as above.)

Round 2

Reviewer 1 Report

The authors have sufficiently addressed my concerns. There are still a few awkwardly worded sentences that could be improved with English writing editing.

Reviewer 2 Report

The manuscript was accordingly revised. Thus it is now considered as acceptable for publication in "Viruses".